# Prediction of the Potential Distribution of the Endangered Species *Meconopsis punicea Maxim* under Future Climate Change Based on Four Species Distribution Models

**DOI:** 10.3390/plants12061376

**Published:** 2023-03-20

**Authors:** Hao-Tian Zhang, Wen-Ting Wang

**Affiliations:** School of Mathematics and Computer Science, Northwest Minzu University, Lanzhou 730030, China

**Keywords:** climate change, endangered plant, potential distribution, species distribution models

## Abstract

Climate change increases the extinction risk of species, and studying the impact of climate change on endangered species is of great significance to biodiversity conservation. In this study, the endangered plant *Meconopsis punicea Maxim* (*M. punicea*) was selected as the research object. Four species distribution models (SDMs): the generalized linear model, the generalized boosted regression tree model, random forest and flexible discriminant analysis were applied to predict the potential distribution of *M. punicea* under current and future climates scenarios. Among them, two emission scenarios of sharing socio-economic pathways (SSPs; i.e., SSP2-4.5 and SSP5-8.5) and two global circulation models (GCMs) were considered for future climate conditions. Our results showed that temperature seasonality, mean temperature of coldest quarter, precipitation seasonality and precipitation of warmest quarter were the most important factors shaping the potential distribution of *M. punicea*. The prediction of the four SDMs consistently indicated that the current potential distribution area of *M. punicea* is concentrated between 29.02° N–39.06° N and 91.40° E–105.89° E. Under future climate change, the potential distribution of *M. punicea* will expand from the southeast to the northwest, and the expansion area under SSP5-8.5 would be wider than that under SSP2-4.5. In addition, there were significant differences in the potential distribution of *M. punicea* predicted by different SDMs, with slight differences caused by GCMs and emission scenarios. Our study suggests using agreement results from different SDMs as the basis for developing conservation strategies to improve reliability.

## 1. Introduction

Climate not only plays an important role in the growth of plants, but also determines the distribution of species [1,2]. Future climate change will shift the distribution of species and even cause the extinction of species [3,4,5,6,7,8,9]. Some studies have indicated that species will disperse to higher latitudes and/or altitudes under climate change characterized by warming [3,10,11,12]. At lower altitudes and latitudes, species richness and diversity are therefore predicted to fall, while at higher altitudes and latitudes, endangered species are predicted to face a greater risk of extinction as a result of increased species competition [13]. Thus, alpine ecosystems in response to climate change will be more sensitive and vulnerable [14,15,16]. Many alpine species, which have ornamental, economic and medicinal values, are at risk from climate change [17,18,19]. In particular, species in the Qinghai-Tibet Plateau have been adversely affected by climate change [20,21,22]. Therefore, understanding the impacts of climate change on the potential distribution of species is of great significance for species conservation.

*Meconopsis punicea Maxim* (*M. punicea*) is a perennial herb belonging to the family *Papaveraceae* and the genus *Meconopsis* [23], mainly distributed in the hillside grassland and alpine shrub at an altitude of 2800–4300 m in northwest Sichuan, northeast Tibet, southeast Qinghai and southwest Gansu, with a flower and fruit period from June to September [12,23,24,25,26]. *M. punicea* is not only a rare resource of Tibetan medicine, but also has a certain ornamental value [23,24]. According to the Red List of Chinese plants database (http://www.chinaplantredlist.org/ (accessed on 12 July 2022)), *M. punicea* is evaluated as a Least Concern (LC) species. Due to global warming, natural ecological degradation and overexploitation of wild resources, the living environment of *M. punicea* has been seriously and repeatedly damaged, and it is facing the danger of exhaustion of natural resources [12,23,27]. Therefore, it is necessary to actively carry out the resource investigation of *M. punicea* and study the response of the potential distribution of *M. punicea* to climate change, which is helpful for the ex situ conservation and the deep resource development of the species, and provides a theoretical basis for the introduction and domestication of the species and resource conservation.

With the development of digital information, species distribution models (SDMs) have been widely used to study the historical geographical distribution of species and their distribution trend under future climate change [28,29]; it provides a reliable theoretical basis for endangered species protection, conservation planning and invasive species control [30,31,32,33,34]. However, there are numerous SDMs algorithms based on statistics and machine learning, and the potential distribution results of species predicted by different algorithms significantly differ [35]. Some studies have shown that being dependent on only one SDM to predict the potential distribution of species would cause the result deviation problem [35,36]. In addition, studies have shown that different global circulation models (GCMs) will bring uncertainty to SDMs prediction [37,38,39], but the uncertainty caused by emission scenarios is significantly higher than that caused by GCMs [37]. Therefore, the reliability of conservation studies for endangered species could be improved by considering the comprehensive results under the influence of various uncertainties, including SDM algorithms, emission scenarios and GCMs.

In this study, we explored the potential distribution of *M. punicea* under climate change using various SDMs (i.e., generalized linear model, generalized boosted regression tree model, random forest and flexible discriminant analysis). We analyzed the agreement of the prediction results of the four SDMs to avoid the uncertainty brought by the SDM algorithm, and also considered the uncertainty brought by emission scenarios and GCMs. Among them, the emission scenarios are based on the scenarios under the sharing socio-economic pathways (SSPs), which provide more diverse air pollutant emission scenarios, and more scientifically describes future climate change under the mode of socio–economic development [40]. Based on the agreement result of the four SDMs, we aimed to determine the potential distribution of *M. punicea* under current and future climate. In addition, we explored changes in the potential distribution of *M. punicea* by comparing potential current and future distributions, and made conservation recommendations based on the potential distribution changes.

## 2. Results

### 2.1. Evaluation of Model Prediction Accuracy and Significance of Bioclimatic Variables

According to the evaluation metric AUC, the performance of the four SDMs were “excellent”, with the GLM having the highest prediction accuracy, followed by the FDA (Figure 1a). According to the evaluation metric Kappa, the prediction accuracy of FDA was the highest, followed by GLM (Figure 1b). Both evaluation metrics consistently showed higher prediction accuracy of the GLM and FDA than that of the RF and GBM (Figure 1). No matter the current or future climate scenarios, the results consistently showed that the SDM algorithm would generate large uncertainties in predicting the potential distribution of *M. punicea* (Figure 2 and Figure 3). The prediction through GLM model showed a large uncertainty in potential distribution caused by the emission scenarios, while under the other SDMs (i.e., GBM, RF and FDA), the uncertainties caused by the emission scenarios were small (Figure 4). Under different GCMs, the potential distributions predicted by the FDA and GLM models consistently showed large uncertainties, while that predicted by the GBM and RF models were less uncertain (Figure 5). In general, the uncertainty generated by GCMs was smaller than that generated by the SDM algorithms and larger than that generated by the emission scenarios.

The importance score of each bioclimatic variable in the four SDMs showed: temperature seasonality (BIO4), mean temperature of coldest quarter (BIO11) and precipitation of warmest quarter (BIO18) have a major contribution to the prediction of the potential distribution of *M. punicea* in the GLM (Table 1). In the GBM and RF, BIO18, precipitation seasonality (BIO15) and BIO11 play a significant role in determining the potential distribution of *M. punicea* (Table 1). In the FDA, the main bioclimatic variables contributing to the potential distribution of *M. punicea* were BIO4, mean diurnal range (BIO2) and isothermality (BIO3) (Table 1). In summary, the dominant bioclimatic variables shaping the potential distribution of *M. punicea* are BIO4, BIO11, BIO15 and BIO18.

Response curves for the dominant bioclimatic variables and potential distribution probabilities fitted through GLM and FDA show similar trends. Among them, the response curves of the distribution probability of *M. punicea* to BIO4, BIO11, BIO15 and BIO18 showed an oscillating trend in a certain range and then stabilized (Figure 6). The response curve trend fitted by GBM and RF were roughly the same, and the response curve of the distribution probability of *M. punicea* to BIO4, BIO11, BIO15 and BIO18 showed a unimodal pattern (Figure 6). The response curves of the four SDMs showed that the temperature seasonality range is 500–750, the mean temperature in the coldest season is below −5 °C, the precipitation seasonality range is above 80 and the precipitation in the warmest season is above 500 mm, which were the most suitable climate environments for the distribution of *M. punicea* (Figure 6). However, the response curve trend of temperature seasonality was different among the four SDMs (Figure 6).

### 2.2. Potential Distribution of M. punicea in Current Climate

The current potential distribution predicted by the four SDMs showed a high consistency, and all the models showed that the potential distribution of *M. punicea* was roughly distributed in Sichuan, Qinghai, Gansu and Qinghai-Tibet (Figure 7). To be specific, the main potential distribution areas were located in Aba Tibetan and Qiang Autonomous Prefecture and Ganzi Tibetan Autonomous Prefecture on the western Sichuan Plateau, Gannan Tibetan Autonomous Prefecture in the south of Gansu Province and Huangnan Tibetan Autonomous Prefecture and Guoluo Tibetan Autonomous Prefecture in the southeast of Qinghai Province, which are concentrated between 29.02° N–39.06° N and 91.40° E–105.89° E (Figure 7). The main potential distribution ranges predicted (29.02° N–39.06° N and 91.40° E–105.89° E) are basically consistent with those of the records (29.53° N–38.32° N and 94.30° E–104.24° E). However, the potential distribution area predicted by each model was slightly different, and the potential distribution area predicted by GLM and FDA was smaller than that predicted by RF and GBM.

### 2.3. Changes in Distribution of M. punicea under Future Climate Change

Under future climate change, the four SDMs all show that the potential distribution area of *M. punicea* expands to the northwest, while a few areas in the eastern part of the current potential distribution area will not be suitable for *M. punicea* in the future (Figure 8). GLM and FDA predicted that by 2050 (2040–2060 average), the potential distribution area of *M. punicea* in the Qinghai-Tibet Plateau would widely extend to northwest China (Figure 8a–d,m–p). However, GBM and RF predicted that the potential distribution area of *M. punicea* will expand less to the northwest by 2050 (Figure 8e–l). In addition, under the SSP5-8.5 scenario (Figure 8b,d,f,h,j,l,n,p), the potential distribution of *M. punicea* by 2050 expanded more to the northwest than that under the SSP2-4.5 scenario (Figure 8a,c,e,g,i,k,m,o). In addition, no matter what the GCM, the main change trend in the potential distribution of *M. punicea* was basically the same. The minor difference was that the loss of potential distribution under the Had was slightly more than that under the BCC (Figure 8).

## 3. Discussion

### 3.1. Comparison of Prediction Results of Four SDMs

In this study, four SDMs were used to predict the potential distribution of *M. punicea*. We compared the prediction performance of the four SDMs by evaluation metrics and the agreement/uncertainty of the model prediction results. The predictive performance of GLM and FDA was better, as judged by the evaluation metrics (AUC and Kappa). In terms of the agreement/uncertainty of the prediction results, the model with good performance did not always perform well. GLM and FDA had the highest agreement and lowest uncertainty under the current climate, contrary to the performance under the SSP2-4.5 scenario and the SSP5-8.5 (Figure 2 and Figure 3). In addition, not all models with good agreement/uncertainty performance had high evaluation metrics (AUC and Kappa). GBM and RF had the highest agreement and lowest uncertainty under the SSP2-4.5 scenario and the SSP5-8.5, while the AUC (Kappa) was relatively low. This mismatch may be caused by different algorithms based on different SDMs. Because the four SDMs selected in this study were carried out under the same environmental variables, as well as the environmental variables selected and the occurrence data were of the same spatial resolution, the only difference lay in the different algorithms used based on different SDMs. Specifically, GLM and FDA are based on generalized linear methods, and GBM and RF are based on classification methods. In addition, emission scenarios generated less uncertainty in predicting potential distributions than GCMs (Figure 4 and Figure 5). Our findings were inconsistent with previous studies in that the uncertainty caused by emission scenarios was greater than that caused by GCM [37,38,39]. The reason for the difference may be the different emission scenarios chosen. Therefore, the reliability of the model is still insufficient when only using model evaluation metrics, and the reliability of the model can be evaluated by calculating the agreement/uncertainty of the predicted results of various models under different emission scenarios and GCMs.

### 3.2. Significant Variables Affecting the Distribution of M. punicea

The potential distribution of plants is impacted by environmental variables, which play a crucial role in plant growth [41], especially precipitation and temperature [42,43]. In this study, the results of the four SDMs consistently showed that the potential distribution of *M. punicea* was significantly impacted by two temperature-related bioclimatic variables (i.e., temperature seasonality (BIO4) and mean temperature of coldest quarter (BIO11)) and two precipitation-related bioclimatic variables (i.e., precipitation seasonality (BIO15) and precipitation of the warmest quarter (BIO18)). *M. punicea* is a type of perennial plant with a seasonal growth and dormancy cycle [23,44,45], and thus strongly correlated with temperature seasonality and precipitation seasonality. The warmest quarter on the Qinghai-Tibet Plateau is usually from July to September, which is also the flowering and fruiting period of *M. punicea* [23,24,26]. Plants in the flowering and fruiting period need proper precipitation; too little precipitation affects the blossom and fruit, and too much water can lead to flower and fruit drops. This is consistent with our results in that the potential distribution probability of *M. punicea* increases with an increase in precipitation of the warmest quarter, and that the optimal water requirement of *M. punicea* in the warmest quarter is when precipitation reaches about 600 mm. In addition, our results show that the increase in the mean temperature of coldest quarter will lead to a decrease in the potential distribution probability of *M. punicea*. This is probably due to the coldest quarter for the dormancy period of *M. punicea*, during which a temperature rise will affect the sprouting and growing during the following year [45,46]. The optimum temperature in the dormancy period of *M. punicea* is below −10 °C, which is consistent with the cold resistance of *M. punicea* [26].

### 3.3. Impacts of Climate Change on the Potential Distribution of M. Punicea

The prediction results of the four SDMs all showed that the potential distribution of *M. punicea* ranged between 29.02° N–39.06° N and 91.40° E–105.89° E under current climate scenarios. Under future climate change, the potential distribution areas of *M. punicea* will be expanded, showing a trend of extending from southeast to northwest. A few areas in the eastern part of the current potential distribution area will not be suitable for *M. punicea* in the future. Our findings are consistent with Zhao et al. [47] in that climate change will expand the potential distribution of *M. punicea*. However, Shi et al. [19] and He et al. [12] suggested that climate change will reduce the potential distribution of *M. punicea*. The reason for the difference may be the different future global climate model chosen or different environmental variables chosen. Instead of representative concentration pathways (RCPs), we selected a Shared Socio-economic Pathways (SSPs) scenario, which is more suitable for China [48,49]. Furthermore, we only selected climate variables, and did not select soil type variables, which may also have an impact on the distribution of *M. punicea*. Compared with RCPs, the SSPs scenario provides more diverse air pollutant emission scenarios, and more scientifically describes future climate change under the mode of socio–economic development.

Our study supports that plants will migrate to higher latitudes under future climate change [3,7], but the potential distribution area of species will not necessarily shrink under future climate change [7,8]. By comparing different climate scenarios in the future, it is found that the potential distribution area of *M. punicea* will expand to the northwest more widely under the SSP5-8.5 scenario. At the end of the 21st Century, the temperature will increase by 4.7–5.1 °C under the SSP5-8.5 scenario, while the temperature will increase by 3.8–4.2 °C under the SSP2-4.5 scenario [50]. The temperature increases more under the SSP5-8.5 scenario, which forces the migration of the cold resistant plant *M. punicea* to higher latitudes. At the same time, it faces greater competition during migration to higher latitudes.

Although our predictions indicate that the potential distribution of *M. punicea* may expand under climate change, the actual movement of species in a changing climate may be characterized by many challenges, such as competition, predation, physical barriers and lack of dispersal media [51,52]. *M. punicea* are entomophilous plants, which mainly rely on flies to transmit pollen [53]. Therefore, a loss of the dispersal potential of flying insects [54,55] may lead to disruption of the transmission of *M. punicea* during migration to higher elevations areas under climate change. In addition, based on the dispersal distance of plants not exceeding 100 m per year [56,57,58,59], we estimate that the species will disperse by 7 km at most in the next 70 years. The potential distribution in the future, expanding by about 120 km (Figure 8), and the dispersal distance of the species is much smaller than the potential distribution in the future. Therefore, most of the potential distribution in the future is unreachable, and only a small part (at most 6%) can be occupied.

### 3.4. Protection Strategies for M. punicea

*M. punicea*, as an endangered plant, is not only of medicinal value, but also of ornamental value [23,24,60]. The potential distribution of *M. punicea* provides a prerequisite for the development of conservation strategies. According to the predictions of various SDMs, the potential distribution area of *M. punicea* will expand to the northwest, while a few areas in the eastern part of the current potential distribution area will not be suitable for *M. punicea* in the future. For places where potential distribution is stable in the future, in situ conservation can be adopted, such as the establishment of nature reserves [19]. In addition, field investigation and assessment should be carried out for places where the potential distribution is gained in the future, and ex situ conservation should be considered [61]. In addition, the habitat of *M. punicea* is being destroyed at an accelerated rate due to overexploitation and climate change [62]. Therefore, it is particularly important to reduce human activities that lead to the loss of biodiversity, strengthen the construction of protected areas and take measures to protect and conserve species in natural habitats.

This study only considered the influence of climate conditions on the suitable areas of *M. punicea*, and did not involve the influence of human activities, terrain and soil, etc. Therefore, the future protection status of this plant needs to be studied in many aspects. In addition, only two future climate scenarios were considered in this study, more climate models and emission scenarios were selected for simulation in subsequent studies, and compared with this study, so as to evaluate the response of *M. punicea*, an endangered medicinal plant, to climate change in a more objective and comprehensive way.

## 4. Materials and Methods

### 4.1. Overview

This study took *M. punicea* as the research object and three primary steps were conducted to implement this study. Firstly, a pairwise Pearson’s correlation test was used to select bioclimatic variables. Secondly, the generalized linear model (GLM), the generalized boosted model (GBM), random forests (RF) and flexible discriminant analysis (FDA) were used to predict the current and future potential distribution of *M. punicea*. At the same time, we considered two global circulation models (GCMs) and two emission scenarios of sharing socio-economic pathways (SSPs) for future climate conditions. Thirdly, the differences in the prediction of four SDMs and the potential distribution of *M. punicea* in response to climate change were analyzed, and different conservation strategies were discussed.

### 4.2. Species Occurrence Data

A total of 230 records of *M. punicea* (i.e., longitude and latitude of the samples) were obtained from the Chinese Virtual Herbarium (http://www.cvh.ac.cn/ (accessed on 12 July 2022)), the Global Biodiversity Information Facility (http://www.gbif.org/ (accessed on 12 July 2022)) and the related literature [12,25]. In order to match the current climate data range (1970–2000 average), only records after 1970 were screened. In order to reduce spatial autocorrelation, only one record was retained within 5 km; that is, a circle was drawn with the sample point as the center of the circle and the radius of 5 km, and all points except the center of the circle were deleted. Finally, 113 records of *M. punicea* were obtained, which were mainly distributed in the eastern part of the Qinghai-Tibet Plateau, including southwest Gansu, southeast Qinghai, northwest Sichuan and northeast Tibet (Figure 9). According to the distribution range of the records of *M. punicea*, the study area was defined as 73° E–110° E and 25° N–45° N.

### 4.3. Environmental Variables

Bioclimatic variables of current climate (representative of 1970–2000) and future climate (2050: average of 2041–2060) were taken from the WorldClim database version 2.1 (http://www.worldclim.org/ (accessed on 1 June 2022)) with a resolution of 2.5 arc-min [63]. For the GCMs of future climate, we considered BCC-CSM2-MR (BCC) and HadGEM3-GC31-LL (Had), which have been commonly used in studies predicting the potential future distribution of species in the Qinghai-Tibet Plateau [12,19,48,49,64]. We also adopted two emission scenarios, including SSP2-4.5 and SSP5-8.5, which represent diverse air pollutant emission scenarios and more scientific descriptions of future climate change under the mode of socio–economic development [48,65,66,67]. SSP2-4.5 represents the scenario of moderate social vulnerability and moderate radiation emission (4.5 W/m^2^), while SSP5-8.5 simulates the development path of traditional fossil fuels and belongs to the scenario of extremely high radiation emission (8.5 W/m^2^) [50]. In order to reduce the correlation between bioclimatic variables; the 113 records of *M. punicea* were first used to extract 19 current climate variables corresponding to geospatial data. Then, we used a pairwise Pearson’s correlation test (r) of bioclimatic variables, and eight bioclimatic variables were selected with |r| < 0.7 (Table 2) [68].

### 4.4. Species Distribution Model

#### 4.4.1. Data Preparation

The presence data used to construct SDMs were set as 113 records of *M. punicea*, and the pseudo-absence data were randomly taken from the study area according to three times the amount of presence data (i.e., 113 presence data, 339 pseudo-absence data). Among them, 80% of presence data (pseudo-absence data) were used as the training set, and the remaining 20% were used as the test set. Three climate scenarios (i.e., current, SSP2-4.5 and SSP5-8.5) were cropped according to the scope of the study area, and then stored in stacks, respectively.

#### 4.4.2. Parameter Setting of the Model

The four SDMs were operated in the R environment (version 4.1.1) using the basic packages “mda” (version 0.5–3), “randomForest” (version 4.6-14) and “gbm” (version 2.1.8), as well as the auxiliary package ‘MASS’ (version 7.3–57), “biomod2” (version 3.5.1). In the GLM model, three forms of variables (i.e., linear term, quadratic term, interaction term) were considered, and the optimal model was determined by stepwise regression according to the AIC value. The stepwise regression was set to bidirectional, while other parameters were kept at default values. In the GBM model, in order to determine the optimal regression tree, it is necessary to consider the number of iterations and learning rate (a smaller rate is better, but the number of iterations should be increased), the complexity of the decision tree (i.e., the depth of the tree), the ratio of resampling and the number of cross validations (used to extract the most appropriate number of regression trees). The number of iterations, the depth of the tree, the learning rate, the resampling ratio and the number of cross validations were respectively set to 10,000, three, 0.01, 0.5 and 10. In the RF model, the number of iterations, which is considered to obtain the optimal model, is set to 1000, and the other parameters were selected as default settings. In the FDA model, the model was adjusted by Multivariate Regression Splines, and the remaining parameters were selected for default settings.

### 4.5. Data Analysis

#### 4.5.1. Model Evaluation Metrics

In this study, the area under receiver operating characteristic curve (AUC) and Kappa were used to evaluate the prediction accuracy of the four SDMs. AUC is the value of the area under the receiver operating characteristic (ROC) curve, where the ROC curve plots sensitivity (Se) versus 1−specificity (Sp) across all possible thresholds between 0 and 1. Se represents the proportion of presences correctly predicted, and Sp represents the proportion of absences correctly predicted [69,70]. As AUC is not affected by the diagnostic threshold, it is recognized as the best evaluation index at present [70,71,72]. The evaluation criteria of AUC are: excellent (0.90–1.00); good (0.80–0.90); general (0.70–0.80); worse (0.60–0.70); fail (0.50–0.60) [30,31,70,71]. Kappa is calculated using the following formula: Kappa = (P_0_ − P_e_)/(1 − P_e_), P_0_ = Pr × Se + (1 − Pr) × Sp and P_e_ = −2 × (Se + Sp − 1) × Pr × (1 − Pr) + P_0_, where Pr is the proportion of presences in the dataset [70]. Kappa is between -1 and 1, usually greater than 0. The larger the value, the higher the accuracy of the model [70,73,74]. Each SDM was randomly simulated 20 times and the AUC (Kappa) was obtained for each simulation. Then, the median of the 20 AUC (Kappa) was used to compare the prediction performance of the model.

In order to further compare the prediction performance of the model, the agreement/uncertainty of the potential distribution predicted by each model was calculated as follows. First, the same part of the potential distribution predicted through each SDM was considered as an agreement. Second, the different part of the potential distribution predicted through each SDM was regarded as an uncertainty.

#### 4.5.2. Comparison of Current and Future Potential Distribution Areas

In species conservation practice, information presented in the form of species presence/absence may be more practical than information presented in terms of probability or suitability. Therefore, a threshold is needed to convert probability or suitability data into presence/absence data [75,76]. Furthermore, the choice of threshold through Kappa maximization is popular in predicting species presence [77,78]. In order to compare the current and future (SSP2-4.5 and SSP5-8.5) changes in potential distribution, the predicted value of the potential distribution was converted as follows. For each model, the operation was repeated 20 times, and the value that made the Kappa value maximum was respectively taken and the average was calculated as the threshold of the model. Then, according to the threshold value, the potential distribution probability of the model was converted into 0–1 data, in which those less than the threshold value were denoted 0 (i.e., absence) and those larger than the threshold value were denoted 1 (i.e., presence). Finally, the calculation was made according to the formula: future potential distribution × 2 + 1 + current distribution. In the calculation results, the number 4 indicates current and future species presence, denoted as “stable”. The number 2 indicates only current species presence, denoted as “loss”. The number 3 indicates the species is not currently present, but will be present in the future, denoted as “gain”.

## Figures and Tables

**Figure 1 plants-12-01376-f001:**
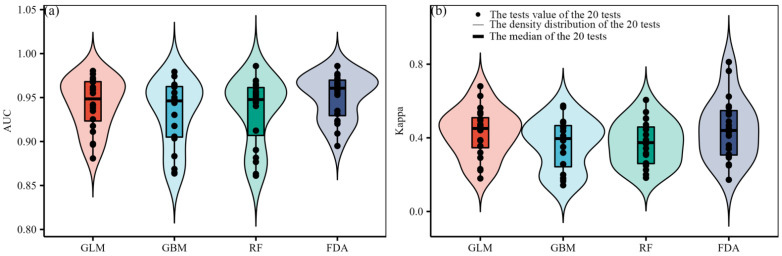
Comparison of prediction accuracy of four species distribution models under (**a**) AUC and (**b**) Kappa. The black dots represent the AUC (Kappa) of each test, the thin outline represents the density distribution of the AUC (Kappa) of 20 random tests, and the thick horizontal line represents the median AUC (Kappa) of 20 random tests.

**Figure 2 plants-12-01376-f002:**
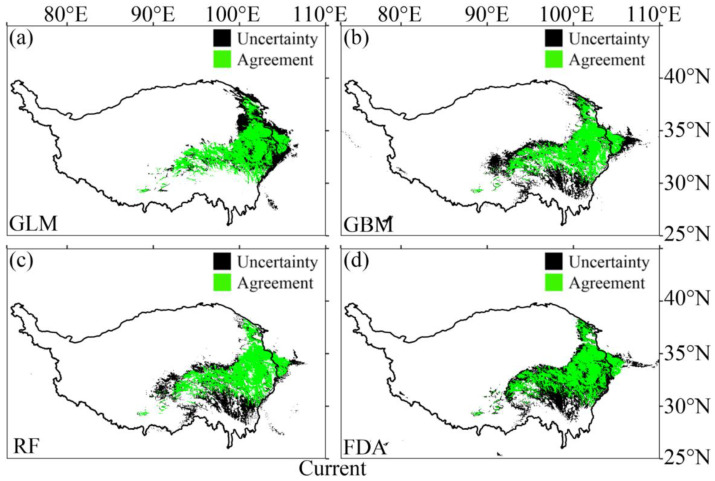
The uncertainty generated by the SDM algorithms in predicting the potential distribution of *M. punicea* under the current climate scenarios. (**a**–**d**) represent the agreement and uncertainty of the potential distribution of *M. punicea* predicted through GLM, GBM, RF and FDA, respectively. The green area indicates the agreement, and the black area indicates the uncertainty. Agreement/Uncertainty means the same/different part of the potential distribution predicted through the four SDMs.

**Figure 3 plants-12-01376-f003:**
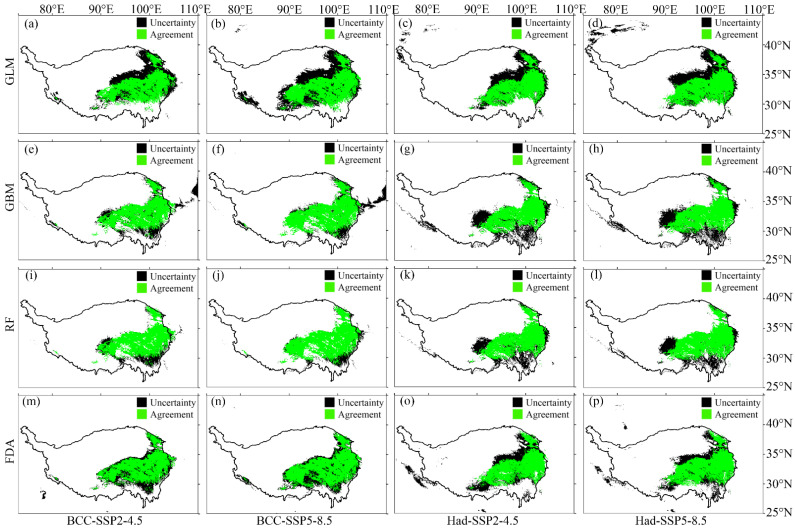
The uncertainty generated by the SDM algorithms in predicting the potential distribution of *M. punicea* under the future climate scenarios. Here, the future global circulation models (GCMs) include BCC-CSM2-MR (BCC) and HadGEM3-GC31-LL (Had), and the emission scenarios include SSP2-4.5 and SSP5-8.5. The first to fourth rows represent the agreement and uncertainty of the predicted potential distribution of *M. punicea* under GLM (**a**–**d**), GBM (**e**–**h**), RF (**i**–**l**) and FDA (**m**–**p**), respectively. The green area indicates the agreement, and the black area indicates the uncertainty. Agreement/Uncertainty means the same/different part of the potential distribution predicted through the four SDMs under the same climate scenarios.

**Figure 4 plants-12-01376-f004:**
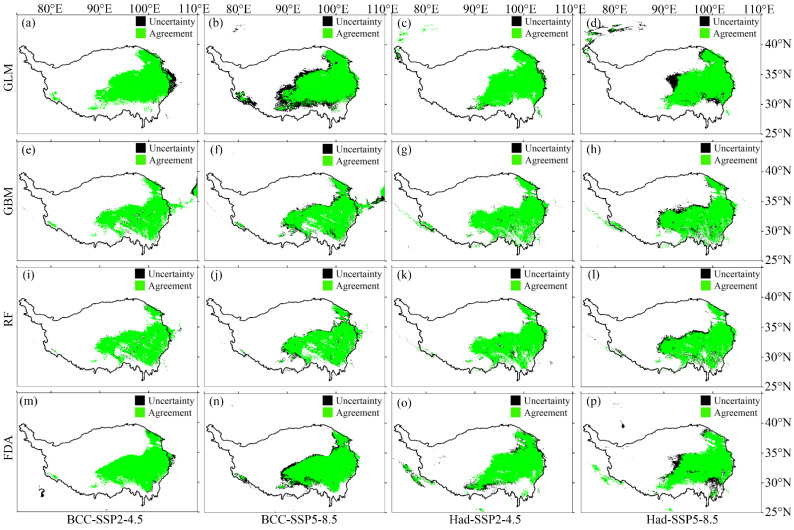
The uncertainty generated by the global circulation models (GCMs) in predicting the future potential distribution of *M. punicea*. Here, the GCMs include BCC-CSM2-MR (BCC) and HadGEM3-GC31-LL (Had), and the emission scenarios include SSP2-4.5 and SSP5-8.5. The first to fourth rows represent the agreement and uncertainty of the predicted potential distribution of *M. punicea* under GLM (**a**–**d**), GBM (**e**–**h**), RF (**i**–**l**) and FDA (**m**–**p**), respectively. The green area indicates the agreement, and the black area indicates the uncertainty. Agreement/Uncertainty means the same/different part of the potential distribution between BCC and Had under the same SDMs and emission scenarios.

**Figure 5 plants-12-01376-f005:**
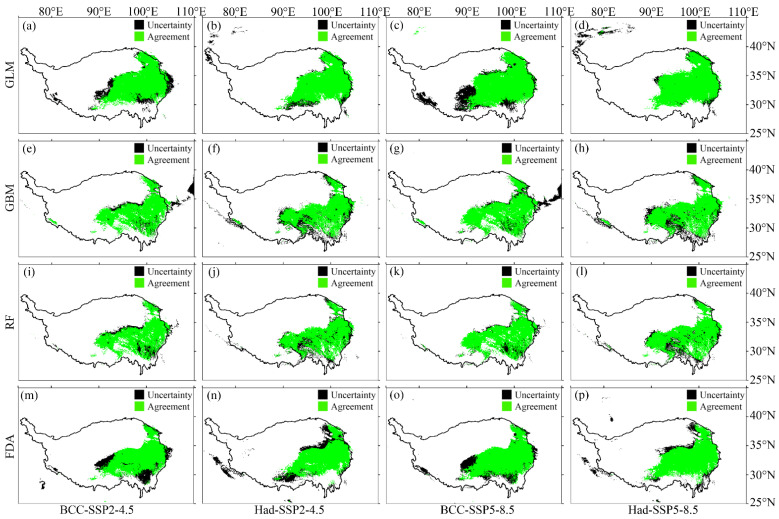
The uncertainty generated by the emission scenarios in predicting the future potential distribution of *M. punicea*. Here, the future global circulation models (GCMs) include BCC-CSM2-MR (BCC) and HadGEM3-GC31-LL (Had), and the emission scenarios include SSP2-4.5 and SSP5-8.5. The first to fourth rows represent the agreement and uncertainty of the predicted potential distribution of *M. punicea* under GLM (**a**–**d**), GBM (**e**–**h**), RF (**i**–**l**) and FDA (**m**–**p**), respectively. The green area indicates the agreement, and the black area indicates the uncertainty. Agreement/Uncertainty means the same part of the potential distribution between SSP2-4.5 and SSP5-8.5 under same SDMs and GCMs.

**Figure 6 plants-12-01376-f006:**
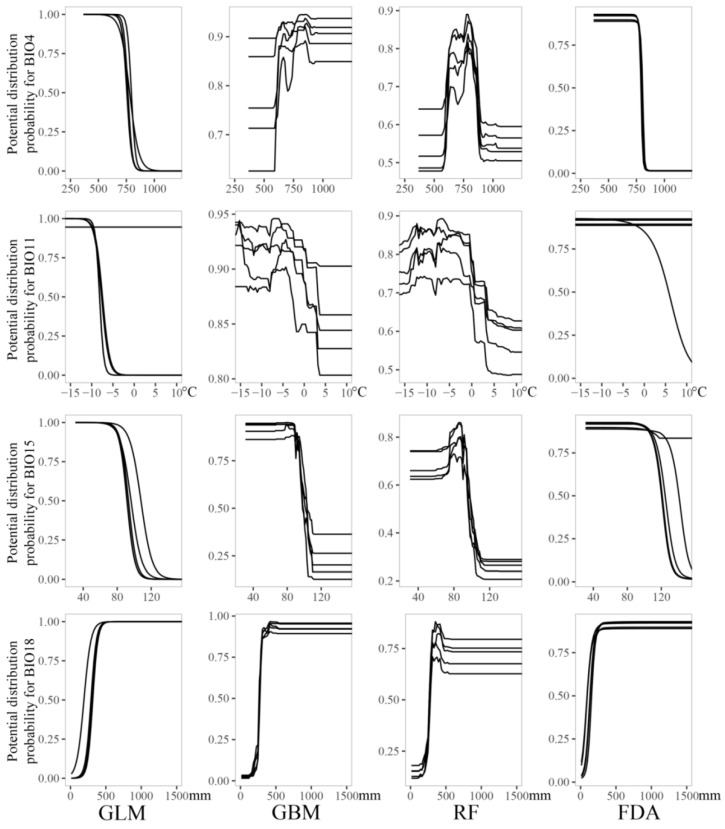
Response curves of potential distribution probability for dominant variables based on four species distribution models for *M. punicea*. The first to the fifth row represent the response curves for BIO4 (temperature seasonality), BIO11 (mean temperature of coldest quarter), BIO15 (precipitation seasonality) and BIO18 (precipitation of warmest quarter), respectively. The first to fourth columns correspond to GLM (generalized linear model), GBM (generalized boosted regression tree model), RF (random forest) and FDA (flexible discriminant analysis), respectively.

**Figure 7 plants-12-01376-f007:**
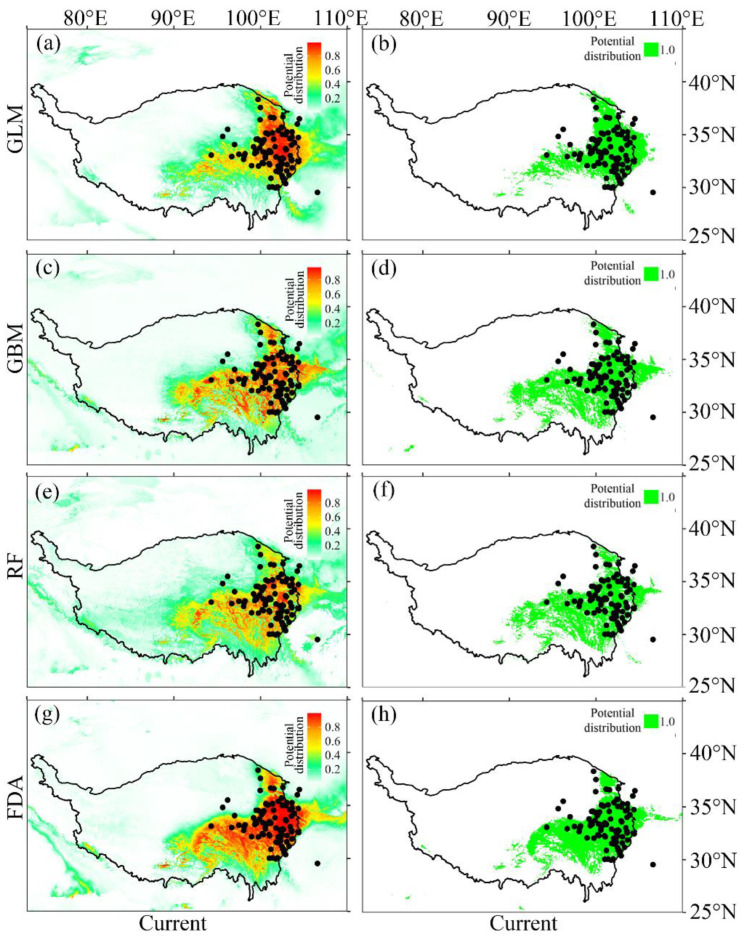
Current potential distribution of *M. punicea* based on four species distribution models. The left column is the prediction of the continuous distribution of *M. punicea* under the current environment. The column on the right shows the 0–1 distribution predictions of *M. punicea* under the current environment. The first to fourth rows represent the predicted potential distribution of *M. punicea* under GLM (**a**,**b**), GBM (**c**,**d**), RF (**e**,**f**) and FDA (**g**,**h**), respectively.

**Figure 8 plants-12-01376-f008:**
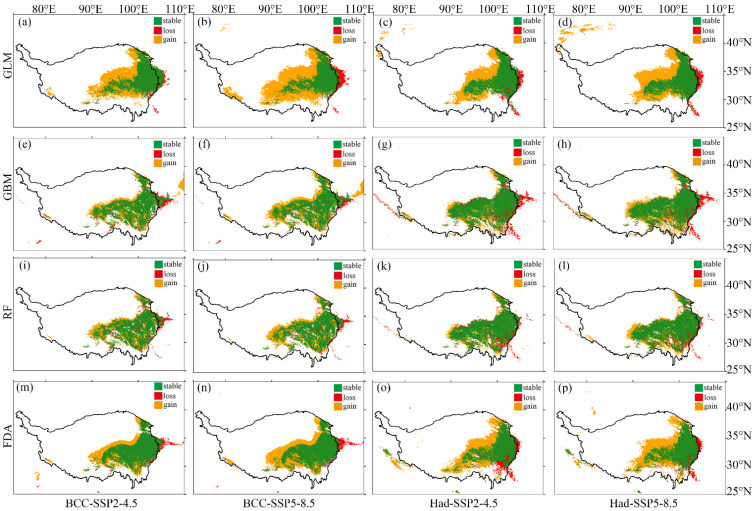
The changes in potential distribution under the future climate scenarios compared to the current climate scenarios based on four species distribution models. Here, the future global circulation models (GCMs) include BCC-CSM2-MR (BCC) and HadGEM3-GC31-LL (Had), and the emission scenarios include SSP2-4.5 and SSP5-8.5. The first to fourth rows show the results of potential distribution changes for GLM (**a**–**d**), GBM (**e**–**h**), RF (**i**–**l**) and FDA (**m**–**p**), respectively. The green area indicates the current and future species presence, denoted as “stable”. The red area indicates the current species presence, but with no future presence, denoted as “loss”. The orange area indicates no current presence, but with presence in the future, denoted as “gain”.

**Figure 9 plants-12-01376-f009:**
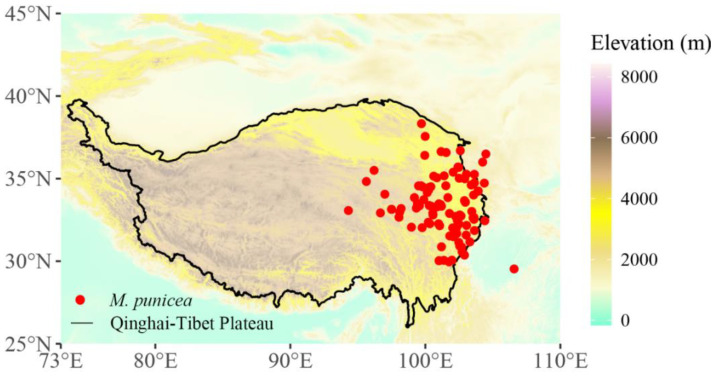
Distribution of occurrence record points of *M. punicea*. The red dots in the figure represent the occurrence record points of *M. punicea*. The polygon data of the Qinghai-Tibet Plateau were obtained through the Global Change Research Data Publishing and Repository (http://www.geodoi.ac.cn/ (accessed on 20 July 2022)).

**Table 1 plants-12-01376-t001:** The importance scores of each bioclimatic variable in four species distribution models (SDMs). The top three most important variables in each SDM are bolded.

	GLM	GBM	RF	FDA
BIO2	0.1502	0.0452	0.0198	**0.8284**
BIO3	0.1276	0.0254	0.0194	**0.6464**
BIO4	**0.9842**	0.0092	0.0468	**0.9972**
BIO5	0.5024	0.0282	0.0398	0.0620
BIO11	**0.6760**	**0.1458**	**0.0742**	0.0096
BIO15	0.48186	**0.2244**	**0.1282**	0.1318
BIO17	0.4210	0.0084	0.0244	0.1826
BIO18	**0.7922**	**0.6958**	**0.3162**	0.3958

**Table 2 plants-12-01376-t002:** Bioclimatic variables used for species distribution models to predict the potential future distribution of *M. punicea*. These variables were screened by Pearson’s correlation test.

Bioclimatic Variables	Meaning of Variables
BIO2	Mean Diurnal Range (mean of monthly (max temp-min temp))/°C
BIO3	Isothermality ((BIO2/BIO7) × 100)
BIO4	Temperature Seasonality (standard deviation ×100)
BIO5	Max Temperature of Warmest Month/°C
BIO11	Mean Temperature of Coldest Quarter/°C
BIO15	Precipitation Seasonality (coefficient of variation)
BIO17	Precipitation of Driest Quarter/mm
BIO18	Precipitation of Warmest Quarter/mm

## Data Availability

Not applicable.

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
