# Peer review of "Prediction of the Potential Distribution of the Endangered Species Meconopsis punicea Maxim under Future Climate Change Based on Four Species Distribution Models"

_plants, 2023, doi:10.3390/plants12061376_

Round 1
Reviewer 1 Report
General review
1. The article considers an issue of high concern, and explores a new methodological approach considering other articles recently published.
2. The text needs revision, once some concepts used are not appropriate or are not clear considering the conservation framework. The structure of the article could be changed, once information about materials and methods is available after “Results” and “Discussion”.
3. In terms of methodology, the basic procedures used in species distribution modeling were followed. But information about methodological decisions, namely in terms of models’ calibration, needs to be improved.
4. In terms of results, outputs have good quality and are appropriate. However, the discussion must be revised in order to improve the text and avoid conclusions that are not supported by results. For example, in lines 196 to 198, the authors mention limitations in terms of growth for the organism under study, but the article does not focus on the impacts of environmental changes on phenological aspects.
Detailed aspects to improve:
Introduction
56 - 60 - revise the text to clarify the idea and present it in a more adequate version
62 - confirm the family affiliation of the species
72 - 74 - what do the authors mean by artificial cultivation? Does it mean conservation ex-situ?
97 - “had a great impact” - Bioclimatic variables influence species distribution. So, bioclimatic variables have higher or lower contribution to explain known distribution. To “have impact” is an expression not adequate in this case.
126 - models outputs are referent to potential distribution or environmental suitability - models do not show distribution;
127 - The “main distribution areas” are related to current distribution or with potential distribution? The idea is not clear.
Fig. 3 - The binary outputs have as legend “Presence”. It must be “Potential distribution”, as suggested before.
3. Discussion
- 3.1. “Combined results” suggests that a combined model will be presented, aiming to prepare a model that reduces the uncertainty associated with results variability. However, it is only discussed the use of metrics to assess models accuracy.
175 - the identified mismatch between predictions and known attributes of the species might be pointing to the influence of a biasing factor. That happens when something in the model calibration process needs further verification. Another argument that supports such idea is that 2 algorithms are presenting coincident results. It is only a problem of AUC, it is a limitation of the algorithm, or it is something in the calibration processes that is biasing the results? The lack of environmental variables, or a mismatch for the adequate spatial resolution between available environmental variables and occurrence data for the species. That should be clarified.
196-198 - the article is focused on potential conditions for occurrence, and does not include experiments about the relationships between climate and phenology. The same idea applies to lines 199 - 200.
203-204 - seasonal patterns in terms of phenology are not clearly presented for the species.
4. Materials and Methods
Some methodological options are not clear:
- the strategy used to create the binary output (0/1) for potential distribution
- the percentage of records used to validate models
In Shi et al. (2022) soil type is indicated as a very determining variable to assess M. punicea’s habitat suitability. Such environmental variable is not included in this exercise. Authors must justify why such variable, and other non-climatic variables, are not considered.
Author Response
Reviewer # 1:
- Response to comment: (The text needs revision, once some concepts used are not appropriate or are not clear considering the conservation framework. The structure of the article could be changed, once information about materials and methods is available after “Results” and “Discussion”.)
Response: Thanks to the reviewer's comments. We have revised the manuscript based on the comments. In order to make the manuscript easier to understand, we have adjusted the logic and expression of the manuscript. In addition, we have rewritten the Introduction and Discussion. According to the requirement of the journal of the Plants, our article is arranged in the order of Introduction, Results, Discussion, Materials and Methods.
- Response to comment: (In terms of methodology, the basic procedures used in species distribution modeling were followed. But information about methodological decisions, namely in terms of models’ calibration, needs to be improved.)
Response: Thanks to the honorable reviewer for valuable suggestions for the improvement of our article. In order to improve models' calibration, we first illustrated the construction of species distribution models (SDMs) in more detail. Then, we calculated the agreement/uncertainty of the predicted results of each SDM. In addition, we added analysis and discussion of agreement/uncertainty. The calculation method is detailed in the second paragraph of Section 4.5.1 in Materials and Methods (the yellow highlighted text). The result is shown as Figure 2. For discussion, please see the yellow highlighted text in the first paragraph of Section 3.1.
- Response to comment: (In terms of results, outputs have good quality and are appropriate. However, the discussion must be revised in order to improve the text and avoid conclusions that are not supported by results. For example, in lines 196 to 198, the authors mention limitations in terms of growth for the organism under study, but the article does not focus on the impacts of environmental changes on phenological aspects.)
Response: Thanks to the honorable reviewer for valuable suggestions for the improvement of our article. According to the suggestion, we have revised the Discussion. To better understand the relationships between climate and phenology of M. punicea. By referring to knowledge about the species, we have discussed how changing values of the most important bioclimatic variables can affect the studied species' survival, germination, flowering and fruiting. For details, please see the yellow highlighted text in the first paragraph of Section 3.2 of the Discussion. In addition, we have discussed the dispersal capacity and the chances the species will fill new potential niches. For details, please see the yellow highlighted text in the third paragraph of Section 3.3 of the Discussion.
- Response to comment: (56–60, revise the text to clarify the idea and present it in a more adequate version.)
Response: Thanks to the honorable reviewer for valuable suggestions for the improvement of our article. The text expression was not clarify and inappropriate before, we have rewritten it. For details, please see the yellow highlighted text in the third paragraph of the Introduction.
- Response to comment: (62, confirm the family affiliation of the species.)
Response: We thank the esteemed reviewer for pointing out the deficiencies in our manuscript. By consulting the materials, we know that Meconopsis punicea Maxim belongs to the Papaveraceae family. This is indeed a mistake in the previous article, which has been corrected. For details, please see the yellow highlighted text in the second paragraph of the Introduction.
- Response to comment: (72–74, what do the authors mean by artificial cultivation? Does it mean conservation ex-situ?)
Response: Yes, artificial cultivation here means ex-situ conservation. For clarity, we have changed artificial cultivation to ex-situ conservation. For details, please see the yellow highlighted text in the second paragraph of the Introduction.
- Response to comment: (97, “had a great impact”-Bioclimatic variables influence species distribution. So, bioclimatic variables have higher or lower contribution to explain known distribution. To “have impact” is an expression not adequate in this case.)
Response: Here, "had a great impact" means that these bioclimatic variables play significant roles in determining the potential distribution of the species, that is, they have a major contribution to the prediction of the potential distribution of the species. In this paragraph, we changed all the ambiguous words "had a great impact", "had a great influence" and "had a significant effect". For details, please see the yellow highlighted text in the second paragraph of Section 2.1 of Results.
- Response to comment: (126, models outputs are referent to potential distribution or environmental suitability-models do not show distribution.)
Response: Thanks to the reviewer's comment. We have changed this inaccurate expression to "the potential distribution of M. punicea was roughly distributed in Sichuan, Qinghai, Gansu and Qinghai-Tibet". For details, please see the yellow highlighted text in the first paragraph of Section 2.2 of Results.
- Response to comment: (127, the “main distribution areas” are related to current distribution or with potential distribution? The idea is not clear.)
Response: Here, "main distribution areas" refer to "main potential distribution areas". We changed this unclear expression to "main potential distribution areas". For details, please see the yellow highlighted text in the first paragraph of Section 2.2 of Results.
- Response to comment: (Fig. 3, the binary outputs have as legend “Presence”. It must be “Potential distribution”, as suggested before.)
Response: Thanks again to the meticulous reviewer. The legend "Presence" is really inaccurate, and the legend should be "Potential distribution". We have corrected this in Figure 4. As a supplementary note, we added a new figure, so the original Figure 3 is recorded as Figure 4 in the revised version.
- Response to comment: (3.1. “Combined results” suggests that a combined model will be presented, aiming to prepare a model that reduces the uncertainty associated with results variability. However, it is only discussed the use of metrics to assess models accuracy.)
Response: Thanks to the reviewer’s comments. In general, the weighted average or ensemble method (combined model) can obtain a better prediction result. We are not here to give a combined model, because of the combined model will not be able to complete the embodiment of all the predictions of a single-model, it would weaken the difference between single-model. In order to further compare the prediction performance of the model, we calculated the agreement/uncertainty of the predicted results of each model. The calculation method is detailed in the second paragraph of Section 4.5.1 in Materials and Methods (the yellow highlighted text). The result is shown as Figure 2. In addition, we have discussed this part again. For details, please see the yellow highlighted text in the first paragraph of Section 3.1 of the Discussion.
- Response to comment: (175, the identified mismatch between predictions and known attributes of the species might be pointing to the influence of a biasing factor. That happens when something in the model calibration process needs further verification. Another argument that supports such idea is that 2 algorithms are presenting coincident results. It is only a problem of AUC, it is a limitation of the algorithm, or it is something in the calibration processes that is biasing the results? The lack of environmental variables, or a mismatch for the adequate spatial resolution between available environmental variables and occurrence data for the species. That should be clarified.)
Response: Thanks to the reviewer's comments. We discussed this part of the content again. For details, please see the yellow highlighted text in the first paragraph of Section 3.1 of the Discussion.
- Response to comment: (196–198, the article is focused on potential conditions for occurrence, and does not include experiments about the relationships between climate and phenology. The same idea applies to lines 199–200.)
Response: To better understand the relationships between climate and phenology of M. punicea. By referring to knowledge about the species, we have discussed how changing values of the most important bioclimatic variables can affect the studied species' survival, germination, flowering and fruiting. For details, please see the yellow highlighted text in the first paragraph of Section 3.2 of the Discussion.
- Response to comment: (203–204, seasonal patterns in terms of phenology are not clearly presented for the species.)
Response: We have discussed this part of the content again. For details, please see the yellow highlighted text in the first paragraph of Section 3.2 of the Discussion.
- Response to comment: (Materials and Methods. Some methodological options are not clear:
- the strategy used to create the binary output (0/1) for potential distribution
- the percentage of records used to validate models.)
Response: We detailed the steps and rationale for converting the potential distribution probability into a binary output (0/1). For details, please see the yellow highlighted text in Section 4.5.2 of Materials and Methods. See the second sentence in Section 4.4.1 for the percentage of records used to validate the models (the yellow highlighted text).
- Response to comment: (In Shi et al. (2022) soil type is indicated as a very determining variable to assess M. punicea’s habitat suitability. Such environmental variable is not included in this exercise. Authors must justify why such variable, and other non-climatic variables, are not considered.)
Response: First of all, we agree that soil type is important for M. punicea. Then, we examined how future climate change might affect the potential distribution of M. punicea. When it comes to future changes, future environmental variables need to be used. However, there is no future soil type data at present, so we can only assume that the current and future soil type data are consistent. Therefore, soil type is not considered here.
Reviewer 2 Report
Review of the submitted manuscript entitled Prediction of the Potential Distribution of the Endangered Species Meconopsis Punicea Maxim under Future Climate Change Based on Four Species Distribution Models ID: plants-2186870
This paper presents the results of modeling the potential distribution of a narrowly distributed and endangered species of the mountain herb Meconopsis punicea. Climate change is one of the greatest threats to biodiversity, especially in mountainous areas. Hence, the purpose of the research seems justified. However, the Introduction could better describe the research problem. I need help finding here information about the research hypothesis and what the authors expected.
Information on methodology should be moved from Introduction to Materials and methods.
My biggest objection is to the methodology; see specific comments. I will give more detailed comments when the authors improve the text regarding appropriate methodology explanations and results descriptions. It is also necessary to improve English. In the current version, the text in parts is not very understandable.
Specific comments:
L.45-60: In making such considerations, authors should keep in mind that even more different prediction results may give different GCMs even using the same prediction algorithm, and the results of such studies should have an estimate of model uncertainty, see, e.g. Olszewski et al 2022, Paź-Dyderska e al 2021.
Paź-Dyderska, S., Jagodziński, A. M., & Dyderski, M. (2021). Possible changes in spatial distribution of walnut (Juglans regia L.) in Europe under warming climate. Regional Environmental Change, 21(1), 18. https://doi.org/10.1007/s10113-020-01745-z
Olszewski, P., Dyderski, M., Dylewski, Ł., Bogusch, P., Schmid‑Egger, C., Ljubomirov, T., Zimmermann, D., Divelec, R. Le, Wiśniowski, B., Twerd, L., Pawlikowski, T., Mei, M., Popa, A. F., Szczypek, J., Sparks, T. H., & Puchałka, R. (2022). European beewolf (Philanthus triangulum) will expand its geographic range as a result of climate warming. Regional Environmental Change, 22, 129. https://doi.org/10.1007/s10113-022-01987-z
For the results obtained here, the authors should develop model compatibility maps.
L.62: Poppaceae-> Papaveraceae
L.77-86: This text fragment should be moved to Materials and methods. In addition, the authors wrote that they considered various variants of SSPs but only one GCM. Here I will emphasize for the second time that there are many of them, and each of them gives different modeling results. This makes the results obtained biased.
L. 157-166: Figures do not indicate which SSPs are involved in each estimate. Maps showing model agreement/estimation uncertainty would need to be calculated. With this, the comparison of modeling results of different methods is more speculative.
L.188-189: Again, maps showing model agreement/uncertainty of estimates would need to be calculated.
L.205-211: What about dispersal capacity? What are the chances the species will fill new potential niches in 2040-2060? It seems easier to say where the species will not be than what areas it will occupy in the future. Please discuss this, taking into account the natural dispersal capacity of the species, the age of onset of reproduction, human impact (plant extraction and cultivation), etc.
L.237-239: It sounds very trivial. What do the authors have in mind?
L.280-289: The authors selected the variables for analysis based on a visual assessment of the biplots graph resulting from the PCA analysis. This is, nevertheless a subjective evaluation. Wouldn't it have been better to use correlation values, and select variables that correlate least with all others, assuming some correlation threshold, e.g. r=|0.7| (see e.g., Olszewski et al 2022)?
Remember that these variables relate to individual seasons throughout the year, in which the plant is in different phenological phases. Please discuss how changing values of the most important bioclim variables can affect the studied species' survival, germination, flowering and fruiting in future, etc. With this, research will be more profound, and the relationship between M. punicea biology and climate will be clearer. The knowledge of the species should be more widely referred and study limitations should be described more broadly. It is also necessary to assess the chances of saturation of new areas of potential occurrence till 2040-2060. Will this species really be a winner of climate change? This should significantly improve the quality of the manuscript.
L.301-306: It is not written what R packages were used to analyze the data. They should have been cited in Materials and methods. No rationale was given for the choice of SSps or GCM. The results obtained in this way are biased by choice of these data. Materials and methods should include information on what period the variables used are for future climate. The analysis of the performance on 113 occurrence points is not much. The number of pseudoabsences and what was the extent of the maps used in the study were not given.
L.324-329: Not explained how statistics were calculated for evaluation metrics.
L.335: How threshold values were determined for obtaining presence/absence maps needed to be explained. This is important, especially if the authors compare the analysis results of several algorithms while questioning the usefulness of statistical metrics for models.
Author Response
- Response to comment: (This paper presents the results of modeling the potential distribution of a narrowly distributed and endangered species of the mountain herb Meconopsis punicea. Climate change is one of the greatest threats to biodiversity, especially in mountainous areas. Hence, the purpose of the research seems justified. However, the Introduction could better describe the research problem. I need help finding here information about the research hypothesis and what the authors expected. )
Response: Thanks to the honorable reviewer for valuable suggestions for the improvement of our article. In order to better describe the research problems and research objectives of this paper, we have rewritten the Introduction and adjusted the logic of the Introduction. For details, please see the yellow highlighted text in Introduction.
- Response to comment: (Information on methodology should be moved from Introduction to Materials and methods.)
Response: Thanks to the honorable reviewer for valuable suggestions for the improvement of our article. We have moved this part of content into Section 4.1 of Materials and Methods, and added the purpose of this research here (the fourth paragraph of the Introduction).
- Response to comment: (My biggest objection is to the methodology; see specific comments. I will give more detailed comments when the authors improve the text regarding appropriate methodology explanations and results descriptions. It is also necessary to improve English. In the current version, the text in parts is not very understandable.)
Response: Thanks to the honorable reviewer for valuable suggestions for the improvement of our article. In light of your detailed comments, we have revised the description of the results, the explanation of the methodology, and the discussion. For details, please see the yellow highlighted text in Results, Discussion, and Materials and Methods. In addition, in order to make the manuscript easier to understand, we have adjusted the logic and expression of the manuscript.
- Response to comment: (L.45-60: In making such considerations, authors should keep in mind that even more different prediction results may give different GCMs even using the same prediction algorithm, and the results of such studies should have an estimate of model uncertainty, see, e.g. Olszewski et al.2022, Paź-Dyderska et al 2021.)
Response: We strongly agree with the reviewer's comments that different GCMs may also cause some uncertainty to the results. We added an illustration of the uncertainty caused by different GCMs in the introduction by referring to the study of Olszewski et al. (2022) and Paź-Dyderska et al. (2021). However, some researchers pointed out that BCC-CSM2-MR in the global circulation model was more suitable for China, and the species we selected were endemic to China, so we only chose the climate scenario under BCC-CSM2-MR to predict the potential distribution of species under future climate. The revised content, please see the yellow highlighted text in the third paragraph of the Introduction.
- Response to comment: (L.62: Poppaceae-> Papaveraceae.)
Response: We thank the esteemed reviewer for pointing out the deficiencies in our manuscript. By consulting the materials, we know that Meconopsis punicea Maxim belongs to the Papaveraceae family. This is indeed a mistake in the previous article, which has been corrected. For details, please see the yellow highlighted text in the second paragraph of the Introduction.
- Response to comment: (L.77-86: This text fragment should be moved to Materials and methods. In addition, the authors wrote that they considered various variants of SSPs but only one GCM. Here I will emphasize for the second time that there are many of them, and each of them gives different modeling results. This makes the results obtained biased.)
Response: Thanks to the sincere suggestions of the reviewer, we have moved this part of content into Section 4.1 of Materials and Methods, and added the purpose of this research here (the fourth paragraph of the Introduction). In addition, we deeply understand the reviewer's comments, but the global circulation model (GCM) selected in this study is the BCC-CSM2-MR that is most suitable for China. Here, we aim to study the uncertainty that a single species distribution model will bring when predicting the potential distribution of species. It is necessary to combine the prediction results of multiple species distribution models simultaneously to minimize the uncertainty caused by the selected models, and to provide some recommendations for the conservation of M. punicea based on the prediction results of multiple models. In subsequent work, we will increase the uncertainty brought about by considering different GCMs.
- Response to comment: (L. 157-166: Figures do not indicate which SSPs are involved in each estimate. Maps showing model agreement/estimation uncertainty would need to be calculated. With this, the comparison of modeling results of different methods is more speculative.)
Response: Thank you very much for the reviewer's comments. Figure 4 shows the potential distribution of species in the current environment, without involving different climate scenarios and different climate models. Figure 5 shows the potential distribution under different climate scenarios. In order to improve the readability of Figures 4 and Figure 5, we have modified the figure notes, respectively.
Thanks to the honorable reviewer for valuable suggestions for the improvement of our article. As suggested by the reviewer, we calculated the agreement/uncertainty of the predicted results of each model. The calculation method is detailed in the second paragraph of Section 4.5.1 in Materials and Methods. The result is shown as Figure 2.
- Response to comment: (L.188-189: Again, maps showing model agreement/uncertainty of estimates would need to be calculated.)
Response: As in Comment 4, we performed the calculations using the same method and the results are presented as Figure 2.
- Response to comment: (L.205-211: What about dispersal capacity? What are the chances the species will fill new potential niches in 2040-2060? It seems easier to say where the species will not be than what areas it will occupy in the future. Please discuss this, taking into account the natural dispersal capacity of the species, the age of onset of reproduction, human impact (plant extraction and cultivation), etc.)
Response: Thanks to the honorable reviewer for valuable suggestions for the improvement of our article. Our previous discussion was not profound enough. According to your suggestions, we have discussed the dispersal capacity and the chances the species will fill new potential niches. We conclude that most of the potential distribution in the future is unreachable due to dispersal capacity, and only a small part (at most 6%) can be occupied. For details, please see the yellow highlighted text in the third paragraph of Section 3.3 of the Discussion.
- Response to comment: (L.237-239: It sounds very trivial. What do the authors have in mind?)
Response: The expression does have some trivial, we corrected this. For details, please see the yellow highlighted text in the first paragraph of Section 3.4 of the Discussion.
- Response to comment: (L.280-289: The authors selected the variables for analysis based on a visual assessment of the biplots graph resulting from the PCA analysis. This is, nevertheless a subjective evaluation. Wouldn't it have been better to use correlation values, and select variables that correlate least with all others, assuming some correlation threshold, e.g. r=|0.7| (see e.g., Olszewski et al 2022)?)
Response: We agree with the reviewer that it is intuitive to select environmental variables using Pearson's correlation coefficient, but it is better to select variables using principal component analysis (PCA) than Pearson's correlation coefficient. This is because PCA first projects the original variables onto the main axes (i.e., gets the main two components, which are linear combinations of the original variables and represent the main information about the original variables). Then, variables are selected based on the correlation between the original variable and the two principal components. In this case, the original variable with a high correlation with the principal component can interpret the largest information. In this way, the most important environmental variable can be found and collinearity can be avoided.
- Response to comment: (Remember that these variables relate to individual seasons throughout the year, in which the plant is in different phenological phases. Please discuss how changing values of the most important bioclim variables can affect the studied species' survival, germination, flowering and fruiting in future, etc. With this, research will be more profound, and the relationship between M. punicea biology and climate will be clearer. The knowledge of the species should be more widely referred and study limitations should be described more broadly. It is also necessary to assess the chances of saturation of new areas of potential occurrence till 2040-2060. Will this species really be a winner of climate change? This should significantly improve the quality of the manuscript.)
Response: Thanks to the honorable reviewer for valuable suggestions for the improvement of our article. According to your suggestions, we have discussed the relationships between climate and phenology of M. punicea by referring to knowledge about the species. For details, please see the yellow highlighted text in the first paragraph of Section 3.2 of the Discussion.
In addition, we roughly estimated the chances of saturation of new areas of potential occurrence till 2040-2060. For details, please see the yellow highlighted text in the third paragraph of Section 3.3 of the Discussion.
- Response to comment: (L.301-306: It is not written what R packages were used to analyze the data. They should have been cited in Materials and methods. No rationale was given for the choice of SSPs or GCM. The results obtained in this way are biased by choice of these data. Materials and methods should include information on what period the variables used are for future climate. The analysis of the performance on 113 occurrence points is not much. The number of pseudo-absences and what was the extent of the maps used in the study were not given.)
Response: The R packages used to analyze the data for this study are detailed in the first sentence of Section 4.4.2 of Materials and Methods (the yellow highlighted text). The rationale for choosing GCM and SSPs are detailed in the third and fourth sentences of Section 4.3 of Materials and Methods (the yellow highlighted text). The first sentence in Section 4.3 of Materials and Methods illustrates what period the variables used for future climate (the yellow highlighted text). The penultimate sentence in Section 4.2 of Materials and Methods describes the analysis of the performance on 113 occurrence points (the yellow highlighted text). The first sentence in Section 4.4.1 of Materials and Methods illustrates the number of pseudo-absence data (the yellow highlighted text). The last sentence in Section 4.2 of Materials and Methods illustrates the study area (the yellow highlighted text).
- Response to comment: (L.324-329: Not explained how statistics were calculated for evaluation metrics.)
Response: We added the calculation methods and formulas for the evaluation metrics. For details, please see the yellow highlighted text in the first paragraph of Section 4.5.1 of Materials and Methods.
- Response to comment: (L.335: How threshold values were determined for obtaining presence/absence maps needed to be explained. This is important, especially if the authors compare the analysis results of several algorithms while questioning the usefulness of statistical metrics for models.)
Response: We thank the esteemed reviewer for pointing out the deficiencies in our manuscript. We have added the missing content. Please see the yellow highlighted text in Section 4.5.2 of the Materials and Methods.
Round 2
Reviewer 2 Report
The authors have significantly improved the work and responded to most of my comments reasonably. However, I am not convinced that PCA was necessary to select bioclim variables. A simple Spearman correlation matrix would have been better. It can be assumed in advance that the bioclim variables did not meet the criterion of normality of distribution (a methodological assumption not only of parametric tests but also of PCA analysis). This is common in such data. Also, I consider the limitation of the study to a single GCM to be a shortcoming of this research, given as it does have a weak link between the role of bioclim variables and their changes in the ecology of the studied species.
Certainly, the manuscript should be linguistically revised.
Author Response
Response to comment: (The authors have significantly improved the work and responded to most of my comments reasonably. However, I am not convinced that PCA was necessary to select bioclimatic variables. A simple Spearman correlation matrix would have been better. It can be assumed in advance that the bioclimatic variables did not meet the criterion of normality of distribution (a methodological assumption not only of parametric tests but also of PCA analysis). This is common in such data. Also, I consider the limitation of the study to a single GCM to be a shortcoming of this research, given as it does have a weak link between the role of bioclimatic variables and their changes in the ecology of the studied species. Certainly, the manuscript should be linguistically revised.)
Response: Thanks to the honorable reviewer for valuable suggestions for the improvement of our article. According to the suggestions, we used the Pearson correlation coefficient (|r|<0.7) to select the bioclimatic variables. The selected bioclimatic variables are detailed in Table 2 in the manuscript. Then, we also recalculated the importance scores of the variables. For details, please see the Table 1 in the manuscript. According to the important scores of each variable, we selected the variables that played a major role in the potential distribution of M. punicea, and redrew the response curve of the variables, as shown in Figure 6. Since three additional variables were selected using Pearson's correlation coefficient than using PCA, we reapplied the new variables to the four species distribution models (SDMs). We recalculated the AUC and Kappa based on the prediction results of each SDM, see Figure 1 for details. Then, we recalculated the agreement and uncertainty based on the current and future potential distribution of M. punicea predicted by each SDM, please see Figure 2 and Figure 3 in the manuscript. We also redrew the spatial map of the current potential distribution of M. punicea, please see Figure 7 in the manuscript. We also calculated the changes in the current and future potential distribution of M. punicea, please see Figure 8 in the manuscript.
In addition, we added another global circulation model (GCM), i.e. HadGEM3-GC31-LL. Now, we considered two GCMs and two emission scenarios for future climate conditions. Then, we predicted the potential distribution of M. punicea in future climates. We calculated the uncertainties generated by GCM and emission scenario in predicting the potential distribution of M. punicea respectively, as shown in Figure 4 and Figure 5. We also discussed the uncertainties caused by GCMs and emission scenarios.
Based on the above changes, some parts of the Abstract, Introduction, Materials, Methods, Results and Discussion have been changed. For details, please see the yellow highlighted text in the manuscript.
We tried our best to improve the manuscript, making various further changes in the manuscript. These changes are marked in yellow in revised paper.